# Reinforcement Learning Policy as Macro Regulator Rather than Macro Placer

**Ke Xue**[1,2], **Ruo-Tong Chen**[1,2], **Xi Lin**[1,2], **Yunqi Shi**[1,2],
**Shixiong Kai**[3], **Siyuan Xu**[3], **Chao Qian**[*1,2]
[1]National Key Laboratory for Novel Software Technology, Nanjing University
[2]School of Artificial Intelligence, Nanjing University
[3]Huawei Noah's Ark Lab

## Abstract

In modern chip design, placement aims at placing millions of circuit modules, which is an essential step that significantly influences power, performance, and area (PPA) metrics. Recently, reinforcement learning (RL) has emerged as a promising technique for improving placement quality, especially macro placement. However, current RL-based placement methods suffer from long training times, low generalization ability, and inability to guarantee PPA results. A key issue lies in the problem formulation, i.e., using RL to place from scratch, which results in limits useful information and inaccurate rewards during the training process. In this work, we propose an approach that utilizes RL for the refinement stage, which allows the RL policy to learn how to adjust existing placement layouts, thereby receiving sufficient information for the policy to act and obtain relatively dense and precise rewards. Additionally, we introduce the concept of regularity during training, which is considered an important metric in the chip design industry but is often overlooked in current RL placement methods. We evaluate our approach on the ISPD 2005 and ICCAD 2015 benchmark, comparing the global half-perimeter wirelength and regularity of our proposed method against several competitive approaches. Besides, we test the PPA performance using commercial software, showing that RL as a regulator can achieve significant PPA improvements. Our RL regulator can fine-tune placements from any method and enhance their quality. Our work opens up new possibilities for the application of RL in placement, providing a more effective and efficient approach to optimizing chip design. Our code is available at `https://github.com/lamda-bbo/macro-regulator`.

## 1 Introduction

In the complex and evolving landscape of modern chip design, placement is a pivotal process that significantly influences the power, performance, and area (PPA) metrics of the final chip [21, 22]. A modern chip typically comprises thousands of macros (i.e., individual building blocks such as memories) and millions of standard cells (i.e., smaller basic components like logic gates). The macro placement result provides a fundamental solution for the subsequent processes (e.g., standard cells placement and routing), thus playing an important role [32]. For example, macro placement influences the placement of standard cells, and poor macro placement might make it challenging to place these cells optimally, leading to an unsatisfactory chip performance [33]. Moreover, an inappropriate macro placement can result in macro blockage in the core center, which harms the overall chip performance by causing unwanted effects such as routing congestion, inferior wirelength, and timing performance issues [26].

---

[*]Correspondence to Chao Qian <qianc@nju.edu.cn>

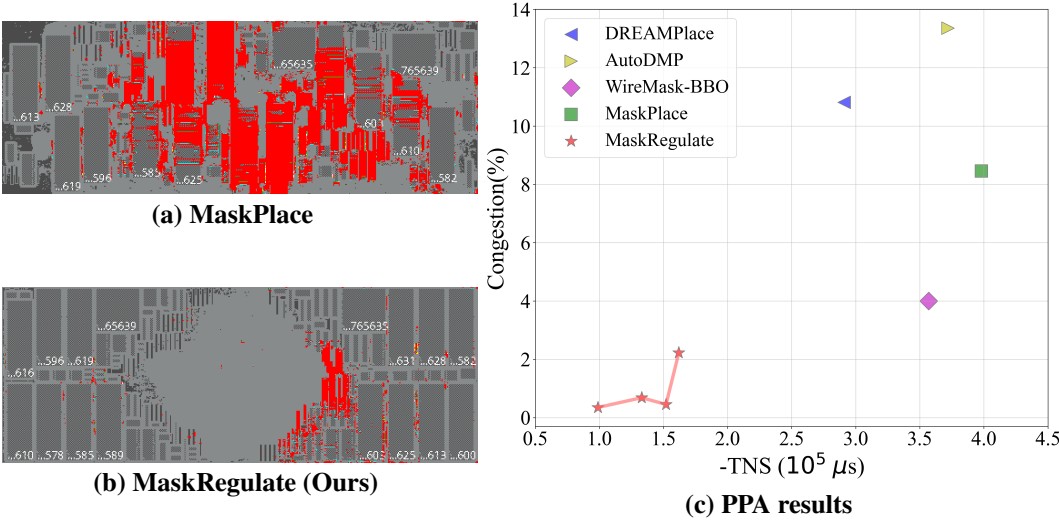

**(a) MaskPlace**

**(b) MaskRegulate (Ours)**

**(c) PPA results**

Figure 1: Placement layouts and congestions of (a) MaskPlace and (b) MaskRegulate on the superblue1 from ICCAD 2015 benchmark [14], where the red points indicate the congestion critical regions. (c): Comparing two crucial PPA metrics, namely Congestion and total negative slack (TNS) between MaskRegulate, DREAMPlace [19], AutoDMP [1], WireMask-EA [29], and MaskPlace [16], where lower values indicate better performance. These results are obtained using *Cadence Innovus*.

Due to the lengthy and complex workflow of chip design, designers often rely on proxy metrics that can reflect the final results to guide the optimization process [2, 30, 20]. One important proxy metric is half-perimeter wirelength (HPWL), which provides an approximation for the routing wirelength and is widely used to measure the placement quality [2, 13, 28]. Traditional macro placement methods can be divided into two categories. Earlier approaches usually solve macro placement by black-box optimization (BBO) [24, 5, 12, 29]. They often suffer from the poor scalability due to the large-scale search space and high complexity of decoding a solution to a placement. Another type is analytical method [6, 7, 19], which can solve the placement efficiently by approximating HPWL gradients. However, these methods are hard to guarantee the non-overlapping constrain between cells and are easy to be stuck in local optima [16, 34].

Reinforcement learning (RL) [31] has recently emerged as a promising technique to enhance the macro placement quality [23, 9, 8, 16, 15]. RL's ability to learn policies through interaction with a complex environment offers a novel pathway for addressing various challenges of macro placement. However, the application of RL is currently hindered by several limitations, including the long training time, an inability to guarantee PPA improvements, and the lack of generalization across different chip layouts. In this work, we highlight that a major contributing factor to these issues is the problem formulation, i.e., the conventional RL approach of placing macros from scratch often results in limited state information and inaccurate reward signal throughout the learning process.

To address these challenges, we propose a novel RL approach called MaskRegulate that shifts the focus from initial placement to refining existing placement layouts. The RL policy acts as a regulator rather than a placer, which operates on pre-existing placements, thus allowing for access to comprehensive state information and enabling the acquisition of more precise rewards. This adjustment enhances the efficiency of the learning process and finally improves the final placement results. Furthermore, MaskRegulate introduces the concept of regularity [26] as a part of input information and a critical reward signal, which has been largely overlooked in previous research despite its significance in ensuring manufacturability and performance. Previous methods often only consider the HPWL metric, suffering from optimizing different metrics effectively. By integrating regularity into the RL framework, our approach aligns more closely with advanced chip requirements.

The effectiveness of the proposed MaskRegulate is comprehensively evaluated on the ICCAD 2015 benchmark [14], which is is currently one of the largest open-source benchmarks that allows us to evaluate PPA metrics such as congestion and timing slack. We first compare the global

HPWL and regularity of our approach against several competitive methods. Additionally, we use the commercial electronic design automation (EDA) tool *Cadence Innovus* to evaluate the PPA performance, demonstrating that our proposed MaskRegulate can lead to significant PPA improvements, e.g., the placement layouts and two PPA metrics on superblue1, as shown in Figure 1. Specifically, compared to MaskPlace (an advanced RL placer [16]; MaskRegulate shares a similar architecture to it), MaskRegulate improves 17.08% on routing wirelength, 73.08% and 38.81 % on routed horizontal and vertical congestion overflow respectively, 18.35% on worst negative slack, 37.89% on total negative slack, and 46.17% on the number of violation points.

This work provides a more effective approach for macro placement of modern chips, opening new possibilities for the application of RL in chip design. The contributions of this work are highlighted in three key points:

- **Novel problem formulation**: Innovatively applying RL in the refinement stage of macro placement, which allows for more effective learning from structured state and accurate reward information, significantly enhancing the learning efficiency and effectiveness.

- **Integration of regularity**: Introducing regularity, a critical yet previously overlooked metric in chip design, into the RL training framework, which not only aligns with industry practice but also enhances the chip PPA quality.

- **Impressive PPA improvement and comprehensive analysis**: On the popular ICCAD 2015 benchmark, our proposed MaskRegulate demonstrates significant improvements in PPA metrics, showing the practical applicability and effectiveness of the RL regulator.

## 2 Background

### 2.1 Placement

The circuit in the placement stage is considered as a graph where vertices model gates. The main input information is the netlist $\mathcal{N} = (V, E)$, where $V$ denotes the information (i.e., height and width) about all macros designated for placement on the chip, and $E$ is a hyper-graph comprised of nets $e_i \in E$, which encompasses multiple cells (including both macros and standard cells) and denotes their inter-connectivity in the routing stage. Given a netlist, a fixed canvas layout and a standard cell library, a placement method is expected to determine the appropriate physical locations of movable macros such that the total wirelength can be minimized. A macro placement solution $\boldsymbol{s} = \{(x_1, y_1), \ldots, (x_k, y_k)\}$ consists of the positions of all the macros $\{v_i\}_{i=1}^{k}$, where $k$ denotes the total number of macros. One popular objective of macro placement is to minimize the total HPWL of all the nets while satisfying the cell density constraint, which is formulated as,

$$\min_{\boldsymbol{s}} HPWL(\boldsymbol{s}) = \min_{\boldsymbol{s}} \sum_{e \in E} HPWL_e(\boldsymbol{s}), \text{ s.t. } D(\boldsymbol{s}) \leq \epsilon, \tag{1}$$

where $D$ denotes the density, $\epsilon$ is a threshold, and $HPWL_e$ is the HPWL of net $e$, which is defined as: $HPWL_e(\boldsymbol{s}) = (\max_{v_i \in e} x_i - \min_{v_i \in e} x_i) + (\max_{v_i \in e} y_i - \min_{v_i \in e} y_i)$.

There are three mainstream placement methods, i.e., analytical methods, black-box optimization methods, and learning-based methods. Analytical methods [4] place macros and standard cells simultaneously, which can be roughly categorized into quadratic placement and nonlinear placement. Quadratic placement [11, 18] iterates between an unconstrained quadratic programming phase to minimize wirelength and a heuristic spreading phase to remove overlaps. Nonlinear placement [6, 20, 7] formulates a nonlinear optimization problem and tries to directly solve it with gradient descent methods. Generally speaking, nonlinear placement can achieve better solution quality, while quadratic placement is more efficient. Recently, there has been extensive attention on GPU-accelerated non-linear placement methods. For example, DREAMPlace [19, 17] transforms the non-linear placement problem in Eq. (1) into a neural network training problem, solves it by classical gradient descent and leverages GPU, enabling ultra-high parallelism and acceleration and producing state-of-the-art analytical placement quality.

Black-box optimization methods for placement have a long history. Earlier methods such as SP [24] and B$^*$-tree [5] have poor scalability due to the rectangular packing formulation. Recently, some black-box optimization methods have made significant progress by changing the search space. AutoDMP [1]

improves DREAMPlace by using Bayesian optimization to explore the configuration space and shows remarkable performance on multiple benchmarks. WireMask-BBO [29] adopts a wire-mask-guided greedy genotype-phenotype mapping and can be equipped with any BBO algorithm, demonstrating the superior performance over other types of methods.

## 2.2 RL for Macro Placement

Researchers recently leverage RL-based methods for better placement quality to meet the demands of modern chip design. GraphPlace [23] first models macro placement as a RL problem. It divides the chip canvas into discrete grids, with each macro assigned discrete coordinates of grids, wherein the agent decides the placement of the current macro at each step. However, no reward is given until all the macros are placed, making the reward sparse and hard to learn. DeepPR [9] and PRNet [8] incorporate macro placement, standard cells placement, and routing to achieve better performance than GraphPlace, but may violate the non-overlap constraint. To address this issue, MaskPlace [16] introduces a dense reward and uses a pixel-level visual representation for circuit modules, which can comprehensively capture the configurations of thousands of pins, enabling fast placement in a full action space on a large canvas size. MaskPlace has many attractive benefits that previous methods do not have, e.g., 0% overlap, dense reward, and high training efficiency. ChiPFormer [15] incorporates an offline learning decision transformer and focuses on improving the generalizability of placer. EfficientPlace [10] integrates a global tree search algorithm to guide the optimization process, achieving remarkable placement quality within a short time.

However, current RL methods exhibit several shortcomings: 1) Placing from scratch provides insufficient state information and inaccurate reward signals; 2) Most methods focus on minimizing wirelength, which may bring macro blockages and thus harm the final PPA metrics. In this work, we propose a novel RL approach for macro placement: an RL policy acts as a macro regulator rather than a macro placer. Specifically, our learned RL policy is designed to adjust macros based on an existing placement result, rather than placing all macros from scratch. This approach aims to refine and optimize pre-existing layouts, addressing the limitations of traditional RL-based placement methods.

# 3 Method

We present our proposed MaskRegulate here. Section 3.1 introduces our problem formulation and policy architecture, and Section 3.2 describes how to integrate regularity into the method.

## 3.1 MaskRegulate Framework

**Problem formulation of RL regulator.** In the Markov Decision Process (MDP) formulation of traditional RL placer, a macro is placed at each step [23, 9, 16, 15]. The placement order of macros is determined based on some pre-defined rules, such as the number of nets, the size of macros, and the number of connected modules that have been placed. An episode ends after all macros have been placed. Typically, the state representation includes information about the chip canvas, the macros that have already been placed, and the macro currently being placed. In GraphPlace [23], the reward is determined only after all macros have been placed, resulting in a sparse reward signal that complicates the training process. Recent works have introduced various methods to densify the reward signal. For instance, WireMask [16] provides a more continuous reward based on the macros already placed. In contrast to RL placers, our RL regulator focuses on refining an existing placement by adjusting the location of one macro at each step. Unlike the placer, which initiates the placement process from scratch, the regulator benefits from additional information when adjusting each macro. Specifically, the regulator considers not only the macros that have already been placed but also the positions of all other macros. Furthermore, it enhances accuracy by taking into account all macros, even while employing a reward function similar to WireMask.

Due to the advantages mentioned above in the MDP problem formulation, even without considering additional factors (e.g., regularity), RL regulator is able to achieve better results compared to RL placer, as shown in our experiments in Appendix B.1. Furthermore, our main experimental results demonstrate superior performance not only in proxy metrics but also in PPA metrics measured by commercial tools, as shown in Section 4.2. The regulator also exhibits better generalization abilities, as shown in Section 4.3. Intuitively, adjusting an unseen chip is easier for the regulator compared to

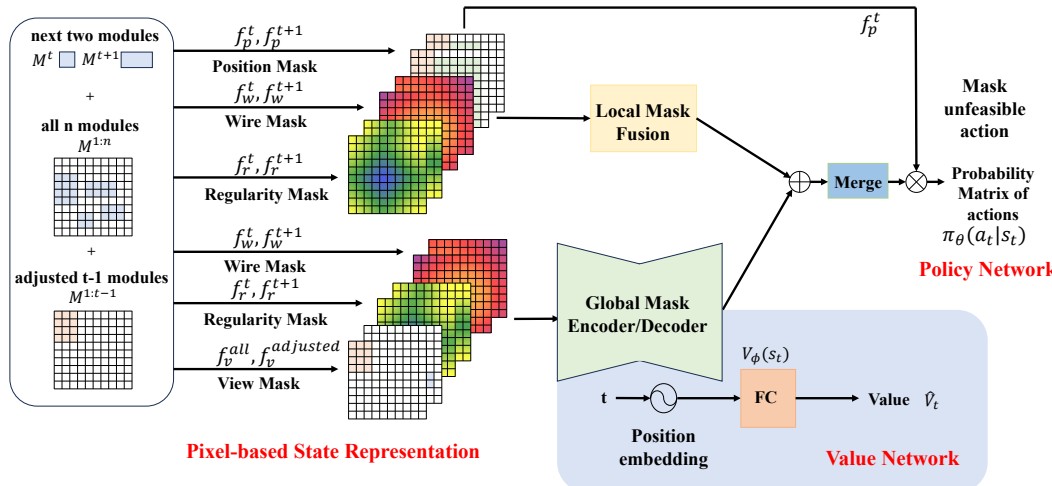

Figure 2: Overview of MaskRegulate. MaskRegulate shares a similar architecture to MaskPlace [16], except for the MDP formulation and the integration of regularity in the state and reward.

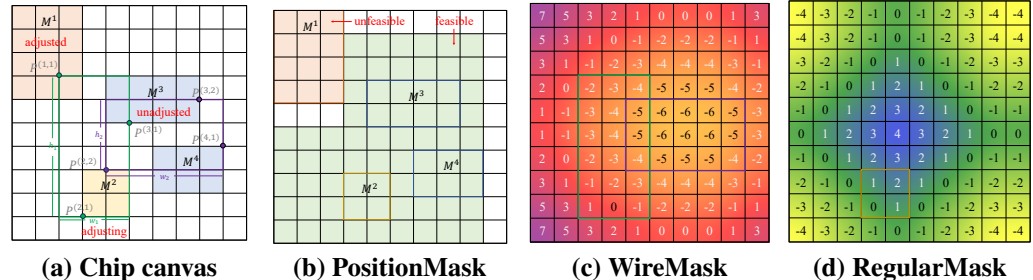

**(a) Chip canvas**  **(b) PositionMask**  **(c) WireMask**  **(d) RegularMask**

Figure 3: Illustration of chip canvas, PositionMask, WireMask and RegularMask. We use the left-bottom corner of the module to denotes its location.

placing macros from scratch, as the incomplete state information of placer would be even worse in the case of unseen chips, resulting in poorer performance.

**Policy architecture.** Our policy architecture is illustrated in Figure 2. The policy divides the chip canvas into several grids and utilizes visual information as inputs, converting chip information into pixel-level image masks. This approach has demonstrated superior efficiency and performance in RL placer policy learning [16, 15]. The inputs include an image of the current canvas, a PositionMask that identifies all valid positions for placing the current macro, a WireMask [16] that indicates the approximate wirelength change for placing the current macro at each valid position, and a RegularMask that indicates the change in regularity for placing the current macro at each valid position (which will be detailed in Section 3.2). An illustration of the PositionMask and WireMask is provided in Figure 3. To facilitate broader adjustments, the PositionMask has been modified to consider only macros that have already been adjusted; thus, grids occupied by unadjusted macros are available for placement. In our MaskRegulate, the calculation of the WireMask is based on all macros, allowing its value to either increase or decrease. These values are normalized to the range $[-1, 1]$, unlike the $[0, 1]$ normalization used in [16]. Additionally, our framework introduces the RegularMask to quantify changes in regularity within the state and to encourage improvements in regularity through the reward function, as presented in Section 3.2.

## 3.2 Integration of Regularity

**Why does regularity matters?** Macro placement has significant impact on subsequent chip design processes, including standard cell placement and routing. If only focusing on minimizing wirelength (which is the case for most current RL placers), certain macros may end up positioned in the middle

of the chip canvas, resulting in macro blockages [26]. This, in turn, leads to the division of available placement areas into separate and disconnected sub-regions. As a consequence, standard cells that are connected by the same net may be scattered across different placement sub-regions, resulting in increased overall wirelength and the potential routing challenges, which ultimately degrade the timing performance. Thus, a well-established practice among experienced engineers in macro placement is to place macros towards the peripheral regions of the chip to prevent macro blockage. In this work, we aim to integrate regularity in the learning-based placement approach to achieve placement preferences similar to those of experienced engineers.

**RegularMask.** Intuitively, macros closer to the edges tend to have lower regularity. Therefore, we propose a simple and effective way to measure regularity. On a canvas, the regularity of a grid located at $(x, y)$ is calculated as $\min\{x, X_{\max} - x\} + \min\{y, Y_{\max} - y\}$, where $X_{\max}$ and $Y_{\max}$ represent the real length of the horizontal and vertical axes, respectively. Given a macro to be placed, the RegularMask measures the value change in regularity for each valid placement position, as illustrated in Figure 3(d).

**Reward and policy learning.** The reward of MaskRegulate consists of two components: $r_{wire}$ and $r_{reg}$, which represent the reduction of HPWL and the improvement in regularity, respectively, after refining the current macro. To mitigate the influence of scale differences on training caused by wirelength and regularity, both $r_{wire}$ and $r_{reg}$ are normalized to $[0, 1]$. The final reward is $r = \alpha \cdot r_{wire} + (1 - \alpha) \cdot r_{reg}$, where $\alpha$ is a trade-off coefficient. We will analyze the influence of $\alpha$ in Section 4.4, showing that different $\alpha$ lead to different multi-objective preferences. The detailed information are presented in Appendix A.3. MaskRegulate treats the chip canvas as a grid and divides it into $N \times N$ cells, resulting in $N^2$ possible discrete actions. We use the popular proximal policy optimization (PPO) algorithm [27] to learn the regulator policy.

## 4 Experiment

In this section, we first introduce the basic experimental settings, including the tasks and evaluation metrics in Section 4.1. Then, we try to answer the following three research questions (RQs) in Sections 4.2 to 4.4: 1) How does MaskRegulate perform compared to other methods? 2) How is the generalization ability of MaskRegulate? 3) How do the different parts of MaskRegulate affect the performance? Finally, we provide the visualization of placement results and congestion in Section 4.5.

### 4.1 Experimental Settings

**Tasks.** We mainly use the ICCAD 2015 benchmark [14] as our test-bed, which includes sufficient advanced chip information and is currently one of the largest open-source benchmarks that allows us to evaluate congestion, timing and other PPA metrics. The benchmark statistics are listed in Table 3 in Appendix A.1. Although ICCAD 2015 is the benchmark we have found that closely reflects the current practices in the EDA industry, it still has some shortcomings. For example, it allows for a large placement area, resulting in loose placement results that do not adhere to the design principles of advanced modern chips. Note that the "A" in PPA denotes "Area", which is a core metric of chip design and should be minimized [3, 32]. Therefore, we scale down the chip's placement area, presenting further challenges for the compared methods. Besides, we also conduct experiments on ISPD 2005 benchmark [25], which is also a popular benchmark in AI for chip design but does not have sufficient information for PPA evaluation. Detailed results can be found in Appendix B.

**Proxy evaluation metrics.** We use the following two popular proxy metrics for a quick comparison of different algorithms: 1) Global HPWL. After determining the locations of all the macros, we use DREAMPlace [19] to place standard cells to obtain the global placement result, and then report the global HPWL (i.e., full HPWL involving both macros and standard cells). Compared to macro HPWL, global HPWL considers the total wirelength, typically on a scale that is two orders of magnitude larger, providing a better estimation of the final real performance of the chip. 2) Regularity: We compute the regularity values for all macros, which serve as a measurement of the overall regularity of the placement result. We run each algorithm for five times and report their mean and variance. We do not consider the rectangular uniform wire density (RUDY) metric [30] for congestion proxy, as this approximation is sometimes positively correlated with the HPWL metric and is not accurate [29]. Instead, we will evaluate congestion within our PPA evaluation.

**PPA evaluation metrics.** The whole chip design process is lengthy and complex, and proxy metrics may not accurately capture the true performance of the chip. PPA metrics often require the use of commercial EDA tools to obtain precise results with expensive cost. In our experiments, we select the best placement result for PPA evaluation based on global HPWL from multiple runs. After obtaining the global placement results, we use commercial tool *Cadence Innovus* to proceed the subsequent stages and evaluate their PPA metrics, including routed wirelength, routed vertical and horizontal congestion overflow, worst negative slack, total negative slack, and the number of violation points. These metrics are extremely important measures of chip design and are typically considered to evaluate the quality of a chip comprehensively.

## 4.2 RQ1: How does MaskRegulate perform compared to other methods?

We consider the following methods to be compared: DREAMPlace [19]: A state-of-the-art analytical placer; AutoDMP [1]: A method that improves DREAMPlace by exploring its configuration space iteratively; WireMask-EA [29]: A state-of-the-art black-box macro placement method with EA as the optimizer; MaskPlace [16]: A representative online RL methods, which shares similar policy architecture, state, HPWL reward with our MaskRegulate.

For the same components, MaskPlace and MaskRegulate use the same settings, e.g., the number of grids, and the learning rate. Detailed information is provided in Appendix A.3. Additionally, in order to demonstrate that the regulator has higher training efficiency than the placer, MaskRegulate and MaskPlace are trained for 1000 and 2000 episodes, respectively. For each chip, MaskRegulate uses DREAMPlace to obtain an initial macro placement result to be adjusted, which takes within few minutes and has relatively low quality.

The overall evaluation results are shown in Table 1. MaskRegulate achieves the best average rank on both proxy and PPA metrics. DREAMPlace has the worst average ranking on wirelength, congestion, and timing. However, after adjustment by MaskRegulate, the obtained placements achieve the best average rank. Compared to MaskPlace, MaskRegulate leads to significant improvements in multiple PPA indicators: improves 17.08% on routing wirelength, 73.08% and 38.81 % on routed horizontal and vertical congestion overflow respectively, 18.35% on worst negative slack, 37.89% on total negative slack, and 46.17% on the number of violation points. By incorporating regularity, MaskRegulate achieves the highest regularity on all the eight chips. We can observe a certain correlation between the proxy metric (global HPWL) and the real metric (rWL), but there still exists a gap, indicating the challenges involved in the placement task. Furthermore, we provide detailed visualizations of placement results in Figure 5, where MaskRegulate shows significant improvements on congestion metrics. Besides, the final placement layouts of MaskRegulate are much regular than all the other methods.

## 4.3 RQ2: How is the generalization ability of MaskRegulate?

The generalization ability of RL policies is an important question to be investigated. In this section, we pre-train MaskRegulate and MaskPlace on the first four chips (i.e., superblue1, superblue3, superblue4, and superblue5) and test on the remaining four chips. To further validate the ability of MaskRegulate to adjust different initial placement results, we use it to adjust the results obtained by different initial placements on the test chips.

The results are shown in Table 2. On both the global HPWL and regularity metrics, MaskRegulate consistently outperforms MaskPlace, showcasing its stronger generalization capability. An interesting finding is that MaskRegulate performs better on unseen chips than on the chips it was trained on, specifically in terms of global HPWL, such as with superblue16. This may suggest that MaskRegulate has learned some general knowledge during the pre-training process, enabling it to overcome local optima that may arise from direct learning on the target chip.

## 4.4 RQ3: How do the different parts of MaskRegulate affect the performance?

We investigate the influence of different parts and provide additional analysis in this section.

**Hyperparameters sensitivity analysis: different trade-off coefficient $\alpha$ leads to different multi-objective preferences.** One hyperparameter of RegularMask is the coefficient $\alpha$ between HPWL reward $r_{wire}$ and regularity reward $r_{reg}$, where a higher $\alpha$ indicates a preference for optimizing

Table 1: Results of proxy metrics and PPA metrics on the ICCAD 2015 benchmarks. Global HPWL (1e8) and Regularity (1e6) are two proxy metrics. PPA metrics are evaluated by *Cadence Innovus*. The placement is performed by different methods, and the subsequent stages are performed by *Cadence Innovus*. rWL (m) is the routed wirelength; rO-H (%) and rO-V (%) represent the routed horizontal and vertical congestion overflow, respectively; WNS (ns) is the worst negative slack; TNS (1e5 $\mu$s) is the total negative slack; NVP (1e4) is the number of violation points. WNS and TNS are the larger the better, while the other metrics are the smaller the better. The best result of each metric on each chip is **bolded**.

| Benchmark | Method | Proxy metrics | | PPA metrics | | | | | |
|---|---|---|---|---|---|---|---|---|---|
| | | Global HPWL | Regularity | rWL | rO-H | rO-V | WNS | TNS | NVP |
| superblue1 | DMP | 8.96 ± 0.84 | 4.15 ± 0.04 | 154.23 | 17.15 | 4.48 | -119.616 | -2.91 | 3.35 |
| | AutoDMP | 8.13 ± 0.17 | 4.99 ± 0.08 | 185.60 | 20.99 | 5.73 | -124.572 | -3.72 | 3.46 |
| | WireMask-EA | 8.07 ± 0.38 | 4.41 ± 0.15 | 149.49 | 7.62 | 0.38 | -67.616 | -3.57 | 2.94 |
| | MaskPlace | 7.93 ± 0.06 | 4.40 ± 0.06 | 158.59 | 16.28 | 0.64 | -72.070 | -3.98 | 4.41 |
| | MaskRegulate | **5.77 ± 0.05** | **3.31 ± 0.00** | **116.11** | **1.26** | **0.11** | **-60.532** | **-1.33** | **1.06** |
| superblue3 | DMP | 12.87 ± 1.73 | 4.43 ± 0.03 | 232.19 | 40.55 | 19.64 | -96.904 | -2.36 | 2.25 |
| | AutoDMP | 8.13 ± 0.69 | 5.49 ± 0.17 | 166.15 | 14.71 | 3.39 | **-76.566** | **-1.12** | 1.44 |
| | WireMask-EA | 9.37 ± 0.81 | 4.77 ± 0.23 | 167.67 | 7.81 | 0.32 | -92.566 | -1.57 | 2 |
| | MaskPlace | 8.90 ± 0.17 | 4.77 ± 0.06 | 177.25 | 9.16 | 0.64 | -111.041 | -1.77 | 2.02 |
| | MaskRegulate | **7.05 ± 0.03** | **3.54 ± 0.00** | **142.89** | **1.86** | **0.18** | -83.635 | -1.15 | **0.97** |
| superblue4 | DMP | 6.81 ± 0.23 | 3.06 ± 0.01 | 132.16 | 20.62 | 4.87 | -73.192 | -1.63 | 2.42 |
| | AutoDMP | 4.57 ± 0.78 | 3.41 ± 0.06 | 82.94 | 5.43 | 0.21 | **-48.137** | **-0.64** | 1.08 |
| | WireMask-EA | 5.51 ± 0.07 | 3.25 ± 0.10 | 110.20 | 8.29 | 0.61 | -83.233 | -1.85 | 1.98 |
| | MaskPlace | 5.28 ± 0.03 | 3.22 ± 0.03 | 106.36 | 9.71 | 0.31 | -67.995 | -1.47 | 1.9 |
| | MaskRegulate | **4.15 ± 0.06** | **2.18 ± 0.02** | **81.78** | **0.29** | **0.11** | -49.071 | -0.90 | **0.88** |
| superblue5 | DMP | 8.78 ± 1.47 | 4.84 ± 0.06 | 144.64 | 3.75 | 0.46 | **-58.907** | **-0.68** | 1.64 |
| | AutoDMP | 12.67 ± 4.09 | 5.79 ± 0.32 | 344.14 | 74.75 | 37.32 | -197.175 | -5.83 | 3.55 |
| | WireMask-EA | 10.23 ± 0.68 | 5.03 ± 0.15 | 189.84 | 4.06 | 0.41 | -75.115 | -1.83 | 2.18 |
| | MaskPlace | 9.81 ± 0.03 | 4.86 ± 0.04 | 196.79 | 4.79 | 0.37 | -118.122 | -2.98 | 2.62 |
| | MaskRegulate | **6.94 ± 0.00** | **4.23 ± 0.01** | **137.79** | **0.02** | **0.02** | -74.83 | -0.73 | **1.32** |
| superblue7 | DMP | 22.70 ± 0.91 | 4.24 ± 0.02 | 427.71 | 100.32 | 73.18 | -123.310 | -6.55 | 7.02 |
| | AutoDMP | 10.04 ± 1.63 | 5.25 ± 0.09 | 221.69 | 6.32 | 0.64 | -52.556 | -1.82 | 4.31 |
| | WireMask-EA | 10.05 ± 0.34 | 4.31 ± 0.13 | 195.36 | 0.82 | 0.35 | -73.070 | -1.78 | 3.45 |
| | MaskPlace | 9.99 ± 0.05 | 4.36 ± 0.03 | 204.32 | 2.77 | **0.33** | -69.441 | -2.18 | 5.78 |
| | MaskRegulate | **7.90 ± 0.03** | **3.03 ± 0.00** | **162.72** | **0.59** | 0.64 | **-50.494** | **-1.48** | **2.27** |
| superblue10 | DMP | 14.81 ± 1.33 | 4.33 ± 0.02 | 261.35 | 7.00 | 5.16 | -83.509 | -3.09 | 2.51 |
| | AutoDMP | 11.48 ± 1.78 | 6.55 ± 0.15 | 234.24 | 2.83 | 1.37 | -169.540 | -2.84 | 1.71 |
| | WireMask-EA | 13.52 ± 2.25 | 4.82 ± 0.12 | 223.08 | 0.76 | 0.55 | -121.785 | -3.11 | **1.65** |
| | MaskPlace | **10.94 ± 0.21** | 4.75 ± 0.11 | 212.85 | **0.51** | 0.19 | -81.916 | -4.37 | 2.11 |
| | MaskRegulate | 11.23 ± 0.38 | **3.90 ± 0.44** | 212.84 | 0.52 | **0.09** | **-77.980** | **-2.74** | 1.75 |
| superblue16 | DMP | 10.22 ± 0.58 | 2.78 ± 0.02 | 187.03 | 74.53 | 30.31 | -138.370 | -4.14 | 4.95 |
| | AutoDMP | 6.12 ± 2.49 | 3.68 ± 0.09 | 119.62 | 2.46 | 0.22 | -41.292 | **-1.17** | 2.26 |
| | WireMask-EA | 6.13 ± 0.14 | 3.17 ± 0.17 | 120.45 | 8.20 | 0.36 | -89.395 | -2.19 | 2.66 |
| | MaskPlace | **5.53 ± 0.04** | 3.08 ± 0.06 | 110.52 | 2.46 | **0.11** | **-36.488** | -1.36 | 2.51 |
| | MaskRegulate | 5.53 ± 0.04 | **2.55 ± 0.12** | 106.82 | 1.40 | 0.17 | -45.962 | -1.77 | **1.88** |
| superblue18 | DMP | 4.97 ± 1.38 | 2.44 ± 0.00 | 85.93 | 11.76 | 8.93 | -73.429 | -0.46 | 0.95 |
| | AutoDMP | 3.02 ± 0.01 | 3.01 ± 0.10 | 61.57 | 1.02 | **0.02** | **-17.545** | **-0.32** | **0.69** |
| | WireMask-EA | 3.46 ± 0.07 | 2.62 ± 0.09 | 69.25 | 1.04 | 0.27 | -34.143 | -0.43 | 1.43 |
| | MaskPlace | 3.49 ± 0.06 | 2.59 ± 0.08 | 70.24 | 1.08 | 0.40 | -43.869 | -0.76 | 1.31 |
| | MaskRegulate | **2.96 ± 0.01** | **1.55 ± 0.00** | **60.64** | **0.04** | 0.03 | -28.285 | -0.40 | 0.85 |
| Average Rank | DMP | 4.625 | 2 | 4.375 | 4.5 | 4.75 | 3.875 | 3.625 | 4.125 |
| | AutoDMP | 3 | 5 | 3.375 | 3.5 | 3.375 | 2.75 | 2.25 | 2.5 |
| | WireMask-EA | 3.75 | 3.75 | 3 | 2.75 | 2.875 | 3.375 | 3.25 | 3 |
| | MaskPlace | 2.375 | 3.25 | 3.125 | 3.125 | 2.375 | 3.125 | 4 | 3.875 |
| | MaskRegulate | **1.25** | **1** | **1.125** | **1.125** | **1.625** | **1.875** | **1.875** | **1.5** |

Table 2: Generalization results of proxy metrics on the four chips of ICCAD 2015 benchmarks. The best result of each metric on each chip is **bolded**.

| Benchmark | MaskRegulate | | MaskPlace | |
|---|---|---|---|---|
| | Global HPWL (1e8) | Regularity (1e6) | Global HPWL (1e8) | Regularity (1e6) |
| superblue7 | **7.99 ± 0.06** | **3.04 ± 0.00** | 10.33 ± 0.17 | 4.24 ± 0.06 |
| superblue10 | **11.55 ± 0.27** | **3.25 ± 0.00** | 11.88 ± 0.72 | 4.73 ± 0.05 |
| superblue16 | **5.16 ± 0.05** | **1.87 ± 0.00** | 5.97 ± 0.20 | 3.11 ± 0.03 |
| superblue18 | **3.04 ± 0.02** | **1.54 ± 0.00** | 3.69 ± 0.09 | 2.52 ± 0.03 |

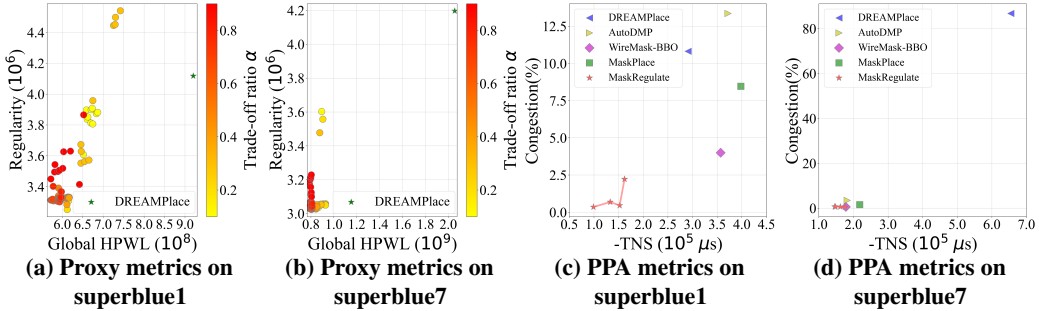

(a) Proxy metrics on superblue1  (b) Proxy metrics on superblue7  (c) PPA metrics on superblue1  (d) PPA metrics on superblue7

Figure 4: Illustration of MaskRegulate regulators with varying $\alpha$ values (ranging from 0.1 to 0.9).

HPWL, and vice versa. In this section, we investigate the influence of the trade-off coefficient $\alpha$. We train different MaskRegulate regulators with varying $\alpha$ values (ranging from 0.1 to 0.9) and report the proxy and PPA results in Figure 4. Due to the expensive computational cost of PPA, we select four different trade-offs of MaskRegulate for evaluation. As expected, different $\alpha$ values lead to different multi-objective preferences. In our experiments, we use $\alpha = 0.7$ for all the chips as it achieves a relative balance between different objectives.

**Ablation studies.** We consider the following ablations of MaskRegulate. 1) Only changing the problem formulation and purely comparing placer and regulator. We implement Vanilla-MaskRegulate, where the only difference to MaskPlace is the problem formulation, and all the other components (e.g., state and reward) are the same. The results show that Vanilla-MaskRegulate consistently outperforms MaskPlace in terms of Global HPWL. 2) MaskRegulate with or without normalization. Since global HPWL has large scale than regularity, MaskRegulate w/o normalization does not prefer to consider regularity, which is not what we expect. 3) Training regularity-aware RL placer from scratch. We implement MaskPlace + RegularMask and compare it with MaskPlace and MaskRegulate. The results show the advantages of the integration of regularity (between MaskPlace and MaskPlace + RegularMask) and our RL regular formulation (between MaskPlace + RegularMask and MaskRegulate). The above ablation results demonstrate the effectiveness of each component of MaskRegulate. Detailed results and discussions are provided in Appendix B.1 due to space limitation.

### 4.5 Visualizations of placement results and congestion.

We provide the detail visualizations of placement results of all the methods on all the eight chips from ICCAD 2015. As shown in Figure 5, our proposed MaskRegulate shows significant improvements on congestion metrics. Besides, the placement result of MaskRegulate is much regular than all the other methods.

### 4.6 Additional results.

We conduct the following additional results to comprehensively show the effectiveness of our MaskRegulate. 1) To verify whether using a better model structure for the RL placer can compare to the regulator, we add comparison with recent proposed ChiPFormer [15] under a fair setting. 2) To further show the generalization ability of our methods, we conduct generalization experiments on the ISPD 2005 benchmark [25]. 3) To investigate whether MaskRegulate can be used to adjust any initial macro placement solution, we use the pre-trained model to fine-tune other placement results.

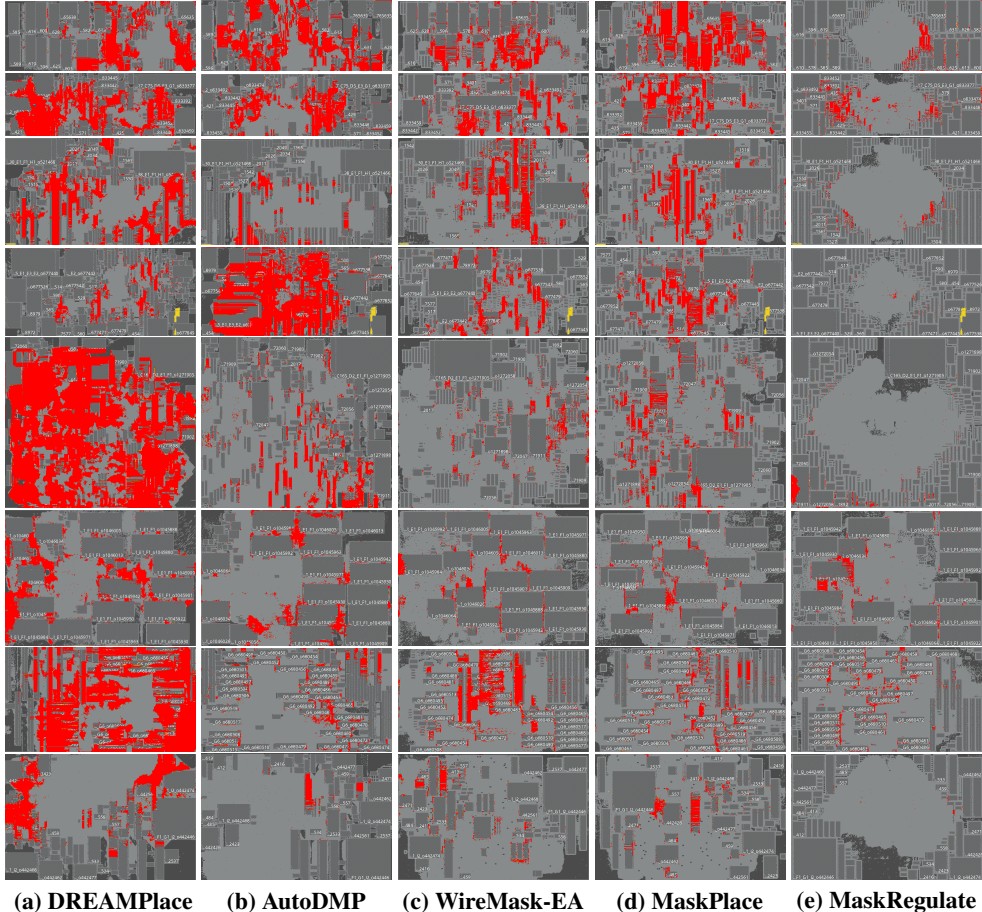

**(a) DREAMPlace**    **(b) AutoDMP**    **(c) WireMask-EA**    **(d) MaskPlace**    **(e) MaskRegulate**

Figure 5: Placement layouts and congestions of different methods on the eight ICCAD 2015 benchmarks. The congestion results are obtained by *Cadence Innovus*, where red points indicate the congestion critical regions.

These results further demonstrate the competitive results of our proposed MaskRegulate. Detailed discussions are provided in Appendix B.2, B.3, and B.4, respectively.

## 5 Final Remarks

**Conclusion.** In this paper, we present a novel RL problem formulation for macro placement, focusing on the development of a macro regulator rather than a placer. Our proposed method, MaskRegulate, demonstrates substantial improvements in chip placement quality by refining existing layouts instead of generating them from scratch. By integrating dense reward signals and emphasizing regularity, our approach effectively addresses the limitations of traditional RL-based placement methods, resulting in superior performance in PPA metrics across various chips. This advancement paves the way for more efficient and effective chip design through RL.

**Limitations and future work.** This study has several primary limitations: it does not consider the impact of module aspect ratio and area factors on placement; it overlooks global wirelength and timing metrics during the training process; and it does not employ advanced transformer architectures [15] to enhance the generalization of the regulator. Chip design inherently involves different preferences, such as the need for compact size in mobile phone chips and larger sizes for computer chips. Therefore, future research should address these challenges and explore efficient methods to obtain a set of chip placements that accommodate different preferences using multi-objective optimization.

## Acknowledgments and Disclosure of Funding

We thank the reviewers for their insightful and valuable comments. This work was supported by the National Science and Technology Major Project (2022ZD0116600), the National Science Foundation of China (62276124), and the Fundamental Research Funds for the Central Universities (14380020).

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

# A  Implementation Details

## A.1  Benchmarks

The detailed statistics of our benchmarks are listed in Table 3.

Table 3: Detailed statistics of the benchmarks.

| Benchmark | #Cells | #Nets | #Pins |
|---|---|---|---|
| adaptec1 | 210,904 | 221,142 | 944,053 |
| adaptec2 | 254,457 | 266,009 | 1,069,482 |
| adaptec3 | 450,927 | 466,758 | 1,875,039 |
| adaptec4 | 494,716 | 515,951 | 1,912,420 |
| bigblue1 | 277,604 | 284,479 | 1,144,691 |
| bigblue2 | 534,782 | 577,235 | 2,122,282 |
| bigblue3 | 1,095,514 | 1,123,170 | 3,833,218 |
| bigblue4 | 2,169,183 | 2,229,886 | 8,900,078 |
| superblue1 | 1,209,716 | 1215710 | 3,767,494 |
| superblue3 | 1,213,253 | 1,224,979 | 3,905,321 |
| superblue4 | 795,645 | 802,513 | 2,497,940 |
| superblue5 | 1,086,888 | 1,100,825 | 3,246,878 |
| superblue7 | 1,931,639 | 1,933,945 | 6,372,094 |
| superblue10 | 1,876,103 | 1,898,119 | 5,560,506 |
| superblue16 | 981,559 | 999,902 | 3,013,268 |
| superblue18 | 768,068 | 771,542 | 2,559,143 |

## A.2  HPWL calculation

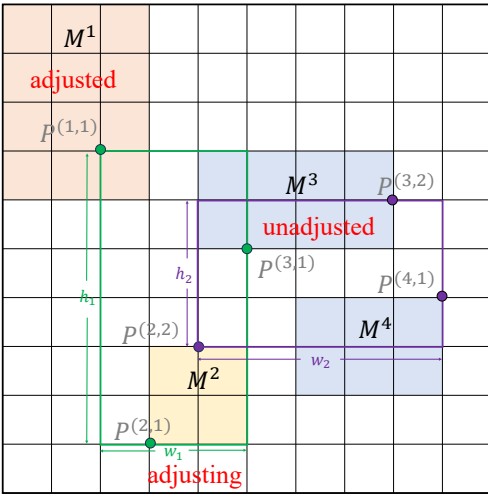

Figure 6: Illustration of chip canvas and calculation of HPWL.

HPWL (half perimeter wirelength) is an important metric which measures the placement quality before routing. Intuitively, Figure 6 illustrates a 2D chip canvas where $M^i$ and $P^{(i,j)}$ denote the $i$-th module to adjust and its $j$-th pin, respectively. Solid boxes in green and purple represent the bounding boxes for two distinct nets on the canvas. In concretely, "Net 1"(in green) connects modules $M^1$, $M^2$ and $M^3$ using wires through pins $P^{(1,1)}$, $P^{(2,1)}$ and $P^{(3,1)}$, while "Net 2"(in purple)

connects modules $M^2$, $M^3$ and $M^4$ using wires through pins $P^{(2,2)}$, $P^{(3,2)}$ and $P^{(4,1)}$. As shown in Figure 3(a), HPWL can be computed as $h_1 + w_1 + h_2 + w_2$.

## A.3   Detailed settings of methods

We conclude some important settings of different methods. For the four compared methods (i.e., DREAMPlace[2], AutoDMP[3], WireMask-EA[4], and MaskPlace[5]), we use their original implementations.

- The size of grids is 224, which is same as the original implementation of MaskPlace [16].

- Hyperparameters.

Table 4: Hyperparameters

| Configuration | Value | Configuration | Value |
|---|---|---|---|
| Optimizer | Adam | Learning rate | $2.5 \times 10^{-3}$ |
| Total episode | 1000 | Epoch for update | 10 |
| Batch size | 64 | Buffer capacity | 5120 |
| Clip $\epsilon$ | 0.2 | Clip gradient norm | 0.5 |
| Reward discount $\gamma$ | 0.95 | Mask soft coefficient | 1 |
| DREAMPlace evaluation number | 3 | Trade-off coefficient $\alpha$ | 0.7 |
| Grid soft coefficient | 4 | | |

- Neural Network architecture.

Table 5: Neural Network architecture

| Block | Layer | Kernel Size | Output Shape |
|---|---|---|---|
| Local Mask Fusion | Conv | $1 \times 1$ | $(224, 224, 12)$ |
| | Conv | $1 \times 1$ | $(224, 224, 12)$ |
| | Conv | $1 \times 1$ | $(224, 224, 1)$ |
| Global Mask Encoder | ResNet-18 | - | 1000 |
| | FC | - | 784 |
| Global Mask Decoder | Deconv | $3 \times 3$ | $(14, 14, 8)$ |
| | Deconv | $3 \times 3$ | $(28, 28, 4)$ |
| | Deconv | $3 \times 3$ | $(56, 56, 2)$ |
| | Deconv | $3 \times 3$ | $(112, 112, 1)$ |
| | Deconv | $3 \times 3$ | $(224, 224, 1)$ |
| Merge | Conv | $1 \times 1$ | $(224, 224, 1)$ |
| Position Embedding | - | - | 64 |
| FC for value | FC | - | 64 |
| | FC | - | 64 |
| | FC | - | 1 |

- Device.
  CPU: Intel(R) Xeon(R) Gold 6430
  GPU: $4 \times$ GeForce RTX 4090

---

[2] https://github.com/limbo018/DREAMPlace

[3] https://github.com/NVlabs/AutoDMP

[4] https://github.com/lamda-bbo/WireMask-BBO

[5] https://github.com/laiyao1/maskplace

# B Additional Experimental Results

## B.1 Ablation studies

**Purely comparison between placer and regulator.** Here, we only change the problem formulation for purely comparing placer and regulator. We implement Vanilla-MaskRegulate, where the only difference to MaskPlace is the problem formulation, and all the other components (e.g., state and reward) are the same. The results in Table 6 clearly demonstrates our motivation, highlighting the advantages of our regulator problem formulation.

Table 6: Results of MaskPlace and Vanilla-MaskRegulate on four chips of ICCAD 2015 benchmarks. The only one difference between these two methods is the problem formulation, where all the other components are the same. The best result on each chip is **bolded**.

| Benchmark | MaskPlace | Vanilla-MaskRegulate |
|---|---|---|
| | Global HPWL (1e8) | |
| superblue1 | $7.93 \pm 0.06$ | $\mathbf{5.58 \pm 0.05}$ |
| superblue3 | $9.00 \pm 0.17$ | $\mathbf{8.01 \pm 0.03}$ |
| superblue4 | $5.28 \pm 0.03$ | $\mathbf{4.31 \pm 0.01}$ |
| superblue5 | $9.81 \pm 0.03$ | $\mathbf{7.89 \pm 0.02}$ |

**MaskRegulate with or without normalization.** The results are shown in Table 7. Since global HPWL has large scale than regularity, MaskRegulate w/o normalization does not prefer to consider regularity, which is not what we expect.

Table 7: Results of MaskRegulate and MaskRegulate without normalization on the four chips of ICCAD 2015 benchmarks. The best result of each metric on each chip is **bolded**.

| Benchmark | MaskRegulate | | MaskRegulate w/o normalization | |
|---|---|---|---|---|
| | Global HPWL (1e8) | Regularity (1e6) | Global HPWL (1e8) | Regularity (1e6) |
| superblue1 | $\mathbf{5.77 \pm 0.05}$ | $\mathbf{3.31 \pm 0.00}$ | $5.82 \pm 0.07$ | $3.42 \pm 0.06$ |
| superblue3 | $7.05 \pm 0.03$ | $\mathbf{3.54 \pm 0.00}$ | $\mathbf{6.71 \pm 0.03}$ | $3.59 \pm 0.01$ |
| superblue4 | $4.15 \pm 0.06$ | $\mathbf{2.18 \pm 0.02}$ | $\mathbf{3.99 \pm 0.02}$ | $2.49 \pm 0.10$ |
| superblue5 | $\mathbf{6.94 \pm 0.00}$ | $\mathbf{4.23 \pm 0.01}$ | $7.03 \pm 0.04$ | $4.26 \pm 0.01$ |

**Training regularity-aware RL placer from scratch.** Our proposed RegularMask and regularity-based reward function can also be used to train a RL placer from scratch. We implement MaskPlace+RegularMask and compare it with MaskPlace and MaskRegulate. The results show the advantages of the integration of regularity (between MaskPlace and MaskPlace + RegularMask) and our RL regular formulation (between MaskPlace + RegularMask and MaskRegulate).

The above ablation results demonstrate the effectiveness of each component of MaskRegulate.

Table 8: Results of MaskPlace, MaskPlace + RegularMask, MaskRegulate on the eight chips of ICCAD 2015 benchmarks. The best result of each metric on each chip is **bolded**.

| Benchmark | MaskPlace | | MaskPlace + RegularMask | | MaskRegulate | |
|---|---|---|---|---|---|---|
| | Global HPWL (1e8) | Regularity (1e6) | Global HPWL (1e8) | Regularity (1e6) | Global HPWL (1e8) | Regularity (1e6) |
| superblue1 | $7.93 \pm 0.06$ | $4.40 \pm 0.06$ | $7.44 \pm 0.08$ | $3.87 \pm 0.06$ | $\mathbf{5.77 \pm 0.05}$ | $\mathbf{3.31 \pm 0.00}$ |
| superblue3 | $8.90 \pm 0.17$ | $4.77 \pm 0.06$ | $7.18 \pm 0.05$ | $\mathbf{3.53 \pm 0.02}$ | $\mathbf{7.05 \pm 0.03}$ | $3.54 \pm 0.00$ |
| superblue4 | $5.28 \pm 0.03$ | $3.22 \pm 0.03$ | $4.49 \pm 0.03$ | $2.21 \pm 0.03$ | $\mathbf{4.15 \pm 0.06}$ | $\mathbf{2.18 \pm 0.02}$ |
| superblue5 | $9.81 \pm 0.03$ | $4.86 \pm 0.04$ | $7.52 \pm 0.08$ | $\mathbf{4.23 \pm 0.02}$ | $\mathbf{6.94 \pm 0.00}$ | $4.23 \pm 0.01$ |
| superblue7 | $9.99 \pm 0.05$ | $4.36 \pm 0.03$ | $8.74 \pm 0.09$ | $3.07 \pm 0.01$ | $\mathbf{7.90 \pm 0.03}$ | $\mathbf{3.03 \pm 0.00}$ |
| superblue10 | $10.94 \pm 0.21$ | $4.75 \pm 0.11$ | $\mathbf{10.90 \pm 0.30}$ | $4.46 \pm 0.26$ | $11.23 \pm 0.38$ | $\mathbf{3.90 \pm 0.44}$ |
| superblue16 | $5.53 \pm 0.04$ | $3.08 \pm 0.06$ | $5.72 \pm 0.03$ | $2.94 \pm 0.06$ | $\mathbf{5.35 \pm 0.05}$ | $\mathbf{1.85 \pm 0.02}$ |
| superblue18 | $3.49 \pm 0.06$ | $2.59 \pm 0.08$ | $2.89 \pm 0.03$ | $1.56 \pm 0.01$ | $\mathbf{2.96 \pm 0.01}$ | $\mathbf{1.55 \pm 0.00}$ |

## B.2  Comparison with ChiPFormer

Recently, ChiPFormer [15] incorporates an offline learning decision transformer to improve the generalizability. However, we find that even after fine-tuning for the same number of episodes as MaskRegulate, it is still challenging to achieve satisfactory results on ICCAD 2015. Due to the resource-intensive nature of fine-tuning it, we conducted only a partial set of experiments, which is why they were not included in the main paper. In the future, we plan to explore training a generalized Regulator based on the transformer and pre-train it on some chips from ICCAD 2015, and compare it with ChiPFormer that pre-train on the same training chips.

Table 9: Results of MaskRegulate and ChiPFormer on eight chips of ICCAD 2015 benchmarks. The best result of each metric on each chip is **bolded**.

| Benchmark | MaskRegulate | | ChiPFormer | |
|---|---|---|---|---|
| | Global HPWL (1e8) | Regularity (1e6) | Global HPWL (1e8) | Regularity (1e6) |
| superblue1 | **5.77 $\pm$ 0.05** | **3.31 $\pm$ 0.00** | 8.09 $\pm$ 0.00 | 4.84 $\pm$ 0.00 |
| superblue3 | **7.05 $\pm$ 0.03** | **3.54 $\pm$ 0.00** | 9.22 $\pm$ 0.02 | 5.13 $\pm$ 0.00 |
| superblue4 | **4.15 $\pm$ 0.06** | **2.18 $\pm$ 0.02** | 5.11 $\pm$ 0.00 | 3.66 $\pm$ 0.00 |
| superblue5 | **6.94 $\pm$ 0.00** | **4.23 $\pm$ 0.01** | 10.97 $\pm$ 0.14 | 5.62 $\pm$ 0.01 |
| superblue7 | **7.90 $\pm$ 0.03** | **3.03 $\pm$ 0.00** | 16.81 $\pm$ 0.09 | 4.78 $\pm$ 0.00 |
| superblue10 | **11.23 $\pm$ 0.38** | **3.90 $\pm$ 0.44** | 14.15 $\pm$ 0.17 | 5.18 $\pm$ 0.00 |
| superblue16 | **5.35 $\pm$ 0.05** | **1.85 $\pm$ 0.02** | 5.88 $\pm$ 0.00 | 3.67 $\pm$ 0.00 |
| superblue18 | **2.96 $\pm$ 0.01** | **1.55 $\pm$ 0.00** | 3.57 $\pm$ 0.00 | 3.18 $\pm$ 0.00 |

## B.3  Experiments on ISPD 2005

We test the generalization on the ISPD 2005 benchmark [25] by directly using the pre-trained models on superblue 1, 3, 4, and 5 (i.e., the same models in Table 2) of MaskPlace and MaskRegulate to place and regulate the eight chips. As shown in Table 10, MaskRegulate still outperforms MaskPlace in most cases, demonstrating our superior generalization ability and robustness.

Table 10: Generalization results of proxy metrics on eight chips of ISPD 2005 benchmarks. The best result of each metric on each chip is **bolded**.

| Benchmark | MaskPlace | | MaskRegulate | |
|---|---|---|---|---|
| | Global HPWL (1e7) | Regularity (1e3) | Global HPWL (1e7) | Regularity (1e3) |
| adaptec1 | 10.58 $\pm$ 0.07 | 4.68 $\pm$ 0.08 | **7.75 $\pm$ 0.12** | **3.33 $\pm$ 0.00** |
| adaptec2 | 13.91 $\pm$ 0.22 | 5.35 $\pm$ 0.03 | **9.53 $\pm$ 0.09** | **4.74 $\pm$ 0.00** |
| adaptec3 | 23.62 $\pm$ 0.25 | 10.04 $\pm$ 0.03 | **20.40 $\pm$ 0.14** | **7.64 $\pm$ 0.01** |
| adaptec4 | **23.92 $\pm$ 0.16** | 11.29 $\pm$ 0.05 | 24.45 $\pm$ 0.06 | **8.60 $\pm$ 0.02** |
| bigblue1 | 10.78 $\pm$ 0.01 | 5.20 $\pm$ 0.04 | **9.34 $\pm$ 0.03** | **2.75 $\pm$ 0.01** |
| bigblue2 | 34.31 $\pm$ 0.31 | **8.92 $\pm$ 0.06** | **24.32 $\pm$ 0.46** | 9.76 $\pm$ 0.01 |
| bigblue3 | 51.53 $\pm$ 0.63 | 10.32 $\pm$ 0.24 | **36.97 $\pm$ 0.35** | **8.08 $\pm$ 0.03** |
| bigblue4 | 134.97 $\pm$ 2.45 | 17.32 $\pm$ 0.20 | **93.96 $\pm$ 0.35** | **12.38 $\pm$ 0.03** |

## B.4  Experiments on fine-tuning existing placement results

To investigate whether MaskRegulate can be used to adjust any initial macro placement solution, we conduct additional experiments to demonstrate this capability. We used the pre-trained model on superblue 1, 3, 4, and 5 (i.e., the same models in Tables 2 and 10) to adjust different placement results obtained by MaskPlace, AutoDMP, and WireMask-EA. The results are shown in Table 11. MaskRegulate consistently improves regularity on all four unseen chips and enhances global HPWL on three chips.

Table 11: Results of proxy metrics on four chips of ICCAD 2015 benchmarks. We use our policy trained on superblue1, superblue3, superblue4 and superblue5 to finetune the placements gained from MaskPlace, AutoDMP and WireMask-BBO on superblue7, superblue10, superblue16 and superblue18. The left column indicates the Global HPWL (1e8) while the right column indicates the regularity (1e6). The best result of each metric on each chip is **bolded**.

| Method | superblue7 | | superblue10 | | superblue16 | | superblue18 | |
|---|---|---|---|---|---|---|---|---|
| MaskPlace | 9.92 | 4.32 | **10.55** | 4.87 | 5.46 | 3.02 | 3.40 | 2.57 |
| MaskRegulate + MaskPlace | **8.53** | **3.05** | 11.20 | **3.25** | **5.27** | **1.89** | **2.88** | **1.56** |
| AutoDMP | 10.68 | 3.96 | 11.57 | 3.73 | 5.87 | 2.56 | 3.02 | 2.29 |
| MaskRegulate + AutoDMP | **8.28** | **3.02** | **11.39** | **3.25** | **5.11** | **1.89** | **2.94** | **1.56** |
| WireMask-EA | 9.53 | 4.33 | **10.99** | 4.86 | 5.91 | 3.10 | 3.37 | 2.60 |
| MaskRegulate + WireMask-EA | **8.54** | **3.02** | 12.02 | **3.25** | **5.20** | **1.85** | **3.01** | **1.54** |

