# OpenReview forum: "Reinforcement Learning Policy as Macro Regulator Rather than Macro Placer"
_NeurIPS.cc/2024/Conference — NeurIPS 2024 poster_

### Official Review · Reviewer_XyiM · 2024-07-11

**Soundness:** 4
**Presentation:** 2
**Contribution:** 4
**Rating:** 6
**Confidence:** 3

**Summary:**

The paper proposes MaskRegulate, a method that utilizes reinforcement learning in the refinement stage for time-efficient training and accurate reward generation. Additionally, the authors introduce a regularity metric that can be used alongside HPWL in chip placement. To validate their proposed method, the authors evaluated the PPA performance, a real-world metric and compared with competitive baselines.

**Strengths:**

- This paper highlights that a major challenge in applying RL to chip placement is the problem formulation.
- The proposed MaskRegulate introduces the concept of regularity, which has been overlooked in previous works, and achieves state-of-the-art performance.
- The authors validated their proposed method by evaluating it on the ICCAD 2015 benchmark and comparing it with competitive baselines. Furthermore, they assessed the real-world metric, PPA, using a commercial EDA tool to ensure the method's practicality in industry applications.
- The visualization results in Figure 6 are impressive and effectively demonstrate the effectiveness of the proposed method. If possible, I recommend including some of these visualizations from Figure 6 in the main paper.

**Weaknesses:**

The overall presentation of the paper requires improvement, particularly in the clarity of labels and explanations for figures. Figure 3(a) is especially problematic, as the labels 'adjusting' and 'adjusted' are ambiguous and do not clearly indicate which blocks they refer to. To avoid potential misinterpretation, it is crucial to enhance both the figure and its accompanying explanation. Addressing these issues will significantly improve the paper's clarity and prevent possible misunderstandings.

**Questions:**

1. How does the model determine adjusted and unadjusted macros? Are the macros adjusted sequentially? If so, how is the order of sequence determined?
2. What happens if the model encounters a situation where no feasible space is available? Does it re-adjust previously adjusted macros?
3. In section 4.4, you mentioned using alpha=0.7 for experiments. Do these experiments refer to the results in Table 1 and Table 2? If so, please explicitly state this, as Figure 4 suggests that alpha is a very important hyperparameter.

**Limitations:**

- Apart from some presentation-related issues, I do not observe any significant limitations in this paper.
- To further strengthen the quality of the work, the authors might consider including training convergence graphs for ChipFormer and MaskRegulate.

---

> ### Author Rebuttal · Authors · 2024-08-07
>
> Thank you for your valuable and constructive comments. Below please find our response.
>
> ### Q1  Figure 3(a) is not clear
> We are sorry for the unclear figures.
> The red macro $M^1$ denotes the macro that has been adjusted; the yellow macro $M^2$ is the current macro that is adjusting; the blue macros $M^3$ and $M^4$ are the macros that are unadjusted and will be adjusted in the future steps. We will revise this figure and accompany explanation in the caption to make it clear. Thank you for bringing this to our attention.
>
> ### Q2 How does the model determine adjusted and unadjusted macros? Are the macros adjusted sequentially?
> Yes, the macros are adjusted sequentially.
> As we stated in lines 145-147 of the paper, the order of the macro sequence is determined by the corresponding net and size of a macro, and the number of connected modules that have been adjusted, which is the same as in previous studies [1-2]. We will revise to make this clear. Thank you.
>
> [1] MaskPlace: Fast Chip Placement via Reinforced Visual Representation Learning. NeurIPS, 2022.
>
> [2] Macro Placement by Wire-mask-guided Black-box Optimization. NeurIPS, 2023.
>
>
> ### Q3 No feasible space is available?
>
> We sincerely appreciate the reviewer's insightful question regarding the model's behavior in situations where no feasible space is available. In fact, throughout our entire training and test process, we did not encounter any instances where the final adjusted solution had overlaps. This can be due to the following three reasons:
>
> 1. Prioritized adjustment sequence. We employ a specific adjustment sequence that prioritizes elements with greater impact on the layout, based on many aspects, as we answered in Q2. This will mitigate the risk of overlap.
> 2. Position mask: MaskRegulate uses a position mask that effectively selects non-overlapping positions, fundamentally reducing the risk of overlapping occurrences.
> 3. Integration of regularity. MaskRegulate is designed to encourage elements to align with edges, which not only enhances the overall layout but also further reduces the probability of overlapping.
>
> Given these strategies, we have not encountered any instances of element overlap during our training and testing phases. However, we acknowledge that the possibility of overlap may exist in extreme scenarios. As you suggested, re-adjusting previously adjusted macros is a good idea for improving the model's robustness, which can lead to legal placement result by additional trials. Besides, we can introduce a negative reward to the overlapping scenario, which would penalize adjustments that lead to overlaps, thereby guiding the model towards better adjustments.
>
> We are grateful to the reviewer for highlighting this potential issue. We will implement this mechanism in the future. This enhancement will further improve our model's robustness and adaptability, enabling it to handle a wider range of complex layout scenarios, including potential space constraint issues. Thank you very much!
>
> ### Q4 Is $\alpha=0.7$ in Table 1 and Table 2?
>
> Yes, we set $\alpha=0.7$ by default in our experiments, including those in Tables 1 and 2. We apologize for not explicitly stating this. We will revise to clarify this setting. Thank you for bringing this to our attention.
>
> ### Further improvement 1: including some visualizations from Figure 6 in the main paper.
>
> Thank you for your appreciation of our visualizations! We're glad that you found them valuable. In the light of your suggestions, we will adjust the layout and attempt to incorporate these visualizations into the revision of our paper. We believe this will enhance readability and improve the overall presentation of our research. Thanks for your constructive feedback.
>
> ### Further improvement 2: adding training convergence graphs
>
> Thank you for your valuable suggestion. We agree that including these figures would provide additional insights and further strengthen the quality of our work. However, due to space limitation of the response phase, we couldn't include these graphs currently. We will include them in the revision of our paper. Thank you.
>
> **We hope that our response has addressed your concerns, but if we missed anything please let us know.**

---

> > ### Comment · Area_Chair_pxxv · 2024-08-11
> > **Dear reviewer, please read and respond to authors' rebuttal.**
> >
> > This paper has diverse reviews and it would benefit a lot to start a discussion to clarify any confusing points.
> >
> > Thanks!
> >
> > Your AC.

---

### Official Review · Reviewer_fJAy · 2024-07-12

**Soundness:** 2
**Presentation:** 1
**Contribution:** 3
**Rating:** 5
**Confidence:** 5

**Summary:**

This paper utilizes an online RL algorithm to adjust the existing placement layouts, instead of placing the blocks on the scratch. Additionally, the paper introduces one heuristic concept regularity, which better regularity results in higher PPA performance. Besides, this paper tests the PPA performance using commercial software, showing their methods can achieve significant PPA improvements. Experiments show the proposed method improves PPA performance and proxy in the ICCAD 2015 benchmark compared to the state-of-the-art placement works.

**Strengths:**

1.	The paper introduces the new metric regularity, which is the heuristic knowledge by experienced engineers. Compared to the proxy metrics commonly used in previous work, it is more reflected in the PPA performance which is a more important metric considered by the industry.
2.	The paper uses the commercial EDA tool to evaluate the efficiency of their methods. The results of the EDA tool analysis are convincing, and the performance is superior to the previous methods.

**Weaknesses:**

Several technical details need to be explained.

1.	The most concerning part is the new MDP. The action space is not well described. Compared to MaskPlace, MaskRegulate places the blocks based on the full placement results. So, if the current placing block is placed in an already occupied position, would the occupied block be switched by the current block, or the occupied block be plugged off the board? Compared to placing the blocks one by one, the termination function is clear that the episode would end once all blocks are placed. In this paper, the termination function is not clear since each step is the full placement with all blocks on the board. What is the termination function in this MDP?
2.	WireMask feature map calculation is also ambiguous in this paper. Since the WireMask feature map is proposed in the MDP that places blocks one by one, it describes the amounts of gains in HPWL when placing the current block on each grid. But, in your MDP, the block seemed to be switched (related to the first question). How do you calculate the WireMask in your case?
3.	In the generalization part, for the unseen chips, is the model training part frozen to do forward inferences only?

**Questions:**

Please see the pros and cons part.

**Limitations:**

The regularity is one heuristic knowledge summarized by experienced engineers. As the introduction in section 3.2 “Why does regularity matters?”, the engineer prefers to place macros toward the peripheral regions of the chip.

Is it possible that there exists some other unknown heuristics, like rectangular shape, or triangle shape that also impact a good performance?

It may be the open question that applying the existing human heuristic knowledge constraints the agent explores the whole state-action design space.

---

> ### Author Rebuttal · Authors · 2024-08-07
>
> Thank you for your valuable and constructive comments. Below please find our response.
>
> ### Q1 Technical details are not clear.
>
> Thank you for carefully reading our paper and providing these detailed and valuable comments. We are sorry for the unclear presentations.
>
> **Q1-1 Termination function.**
>
> MaskRegulate sequentially adjusts all the macros in a certain order. All macros will be sorted according to certain rules and then adjusted one by one, as we stated in line 146 of the main paper. The MDP will terminate when all macros have been adjusted.
>
> **Q1-2 Would the occupied block be switched by the current block?**
>
> No. When MaskRegulate is adjusting a macro, only the macros that have been adjusted (i.e., macros that appear earlier in the adjustment sequence) are considered as occupied, as illustrated in Figure 3(b) of the paper. The macro to be adjusted can temporarily occupy the position of unadjusted macros, and the HPWL is calculated in their respective nets.
>
> We will revise these parts to make them clear. Thank you very much.
>
> ### Q2 Calculation of WireMask
> The WireMask is calculated by all the other macros' current locations, whether they are adjusted or not, as illustrated in Figure 5 of our paper.
>
> ### Q3 Model inference in the generalization part.
> Yes. We frozen the pre-trained model and only do forward inference in the generalization experiments. We will make this clear in our final revision. Thank you.
>
> ### Q4 Integrate unknown heuristics
>
> Thank you for your insightful comment and valuable suggestion. We would like to address your point as follows:
>
> In the chip design domain, heuristic knowledge plays a crucial role, and state-of-the-art chips still heavily rely on expert knowledge. We acknowledge that there may exist other unknown heuristics that could potentially impact the performance of the placement process. Incorporating these additional heuristics is an important direction for future research. In our upcoming work, we plan to explore techniques like reward modeling to effectively capture and integrate a wider range of expert knowledge to guide the training of our RL regulator.
>
> This work represents an initial attempt to incorporate advanced heuristic knowledge, specifically the idea that macros should be placed near the periphery of the chip, into the current practice of RL-based placement. Our results demonstrate that this approach yields significant improvements in placement quality. We believe that this work opens a door to enhance RL-based approaches by incorporating heuristic knowledge to make them more suitable for real-world placement tasks.
>
> We greatly appreciate your suggestion and will include a discussion of these potential future directions in the revised version of our manuscript. Your feedback has been invaluable in helping us identify areas where we can clarify and expand upon our contributions.
>
> **We hope that our response has addressed your concerns, but if we missed anything please let us know.**

---

> > ### Comment · Area_Chair_pxxv · 2024-08-11
> > **Dear reviewer, please read and respond to authors' rebuttal.**
> >
> > This paper has diverse reviews and it would benefit a lot to start a discussion to clarify any confusing points.
> >
> > Thanks!
> >
> > Your AC.

---

### Official Review · Reviewer_KS3m · 2024-07-13

**Soundness:** 3
**Presentation:** 3
**Contribution:** 3
**Rating:** 7
**Confidence:** 4

**Summary:**

This paper proposes a novel approach called MaskRegulate for the refinement stage of macro placement, using reinforcement learning (RL) methods. Specifically, it trains an RL policy to adjust the existing placement layouts, from which the policy can receive sufficient information, instead of placing from scratch. Regularity is also considered during the learning process, helping to enhance the placement quality. Experiments demonstrate that it outperforms previous competitive approaches, achieving significant half-perimeter wirelength (HPWL), power, performance, and area (PPA) improvements.

**Strengths:**

1.	This paper first explores RL method as a regulator instead of a placer, providing a more
effective and efficient approach to optimizing chip design.
2.	Regular mask is employed to guide the learning process, which greatly improves the regularity of the chip layouts.
3.	PPA metrics are considered for a comprehensive analysis, showing the practical applicability and effectiveness of MaskRegulate.
4.	The experiments are sufficient and solid, showing the great refinement and generalization ability of MaskRegulate.

**Weaknesses:**

1.	Regularity is also proposed in other works such as [1]. This paper does not discuss how it differs from them.
2.	The paper lacks some discussion on the necessity of using RL other than other methods as the regulator.

[1] A. Vidal-Obiols, J. Cortadella, J. Petit, M. Galceran-Oms, and F. Martorell. Rtl-aware dataflow-driven macro placement. In 2019 Design, Automation Test in Europe Conference Exhibition (DATE), 2019.

**Questions:**

1.	Can the regular mask be used in [2] WireMask-BBO?
2.	Why does GP HPWL improve since only MP HPWL is considered in the learning process?

[2] Y. Shi, K. Xue, L. Song, and C. Qian. Macro placement by wire-mask-guided black-box optimization. In Advances in Neural Information Processing Systems 36, New Orleans, LA, 2023.

**Limitations:**

Yes

---

> ### Author Rebuttal · Authors · 2024-08-07
>
> Thank you for your valuable and positive comments. Below please find our response.
>
> ### Q1 Discussion of regularity in other paper
>
> Thanks for your valuable comments.
>
> [1] proposes a multi-level approach for macro placement that can leverage the hierarchy tree and effectively explore structural information at the RTL stage. The hierarchical partitions facilitate the integration of information such as dataflow, wirelength, and regularity. Then, [2] attempts to "mimic" the interaction between the RTL designer and the physical design engineer to produce human-quality macro placement results by exploiting the logical hierarchy as well as regularity and connectivity of macros.
>
> However, no studies have integrated regularity into RL. In our paper, we not only propose a new RL regulator framework but also integrate regularity into it. Additionally, our proposed RegularMask can be used to improve other methods, such as MaskPlace (as shown in Table 8 of the paper) and WireMask-BBO (as shown in the following answers to Q2). We will add this discussion to our revised paper. Thank you very much.
>
> [1] RTL-aware dataflow-driven macro placement. DATE, 2019
>
> [2] RTL-MP: Toward practical, human-quality chip planning and macro placement. ISPD, 2022.
>
>
> ### Q2 Discussion of using RL as regulator. Can the regular mask be used in WireMask-BBO?
>
> Thank you for your valuable comments. Compared to RL, BBO relies on iterative evaluation and cannot obtain a generalizable policy capable of rapid adjustment through forward inference alone. We believe other methods (such as BBO) can also utilize our proposed RegularMask. To investigate whether WireMask-BBO can benefit from incorporating RegularMask, we conducted additional experiments on the ICCAD 2015 benchmark to compare WireMask-EA and WireMask-EA + RegularMask. WireMask-EA + RegularMask comprehensively places and adjusts based on both WireMask and RegularMask. As shown in Table R3, WireMask-EA + RegularMask improves both global HPWL and regularity in all the cases, demonstrating the versatility of our proposed RegularMask. We will add this discussion and experiment to our revised paper. Thank you.
>
>
> ### Q3  Why does GP HPWL improve since only MP HPWL is considered in the learning process?
>
> Macros (i.e., individual building blocks such as memories) have much larger sizes than standard cells (i.e., smaller basic components like logic gates), which significantly impacts the overall quality of the final chip. A common practice in placement is to first fix macro locations and then place standard cells [1-3]. After determining the positions of macros by optimizing MP HPWL, placing the remaining standard cells typically yields good final results, i.e., GP HPWL. However, to consistently improve GP HPWL, it's necessary to consider the sparse signal in the GP stage, which is one of our future areas of work as we stated in our paper. We will add this discussion to our revised paper. Thank you very much.
>
> [1] MaskPlace: Fast chip placement via reinforced visual representation learning. NeurIPS, 2022.
>
> [2] AutoDMP: Automated DREAMPlace-based macro placement. ISPD, 2023.
>
> [3] Macro placement by wire-mask-guided black-box optimization. NeurIPS, 2023.
>
> **We hope that our response has addressed your concerns, but if we missed anything please let us know.**

---

> > ### Comment · Reviewer_KS3m · 2024-08-11
> >
> > Thank you very much for addressing my concerns. I will keep my score.

---

### Official Review · Reviewer_NxAR · 2024-07-17

**Soundness:** 3
**Presentation:** 2
**Contribution:** 3
**Rating:** 7
**Confidence:** 5

**Summary:**

This paper introduces a Macro Regulator that uses reinforcement learning (RL) to optimize existing macro placements. It reconsiders the application of RL in macro placement and incorporates a regularity metric into the reward function. The paper presents a comprehensive set of comparative experiments to ultimately demonstrate the superiority of the Macro Regulator. Ablation studies also confirm the effectiveness of each component in the proposed approach.

The contributions of the paper are specified as follows: Firstly, Macro Regulator shifts the focus of RL from placing macros from scratch to refining existing macro placements. This approach enables the RL policy to utilize more state information and achieve more accurate reward signals, enhancing learning efficiency and final PPA quality. Secondly, the paper introduces the concept of regularity into the RL framework, a crucial metric often overlooked in chip design. By incorporating regularity as part of the input information and reward signal, Macro Regulator ensures consistency in manufacturing and performance. Thirdly, the proposed method is evaluated on the ICCAD 2015 benchmark, demonstrating superior performance in terms of global placement half-perimeter wire length (HPWL) and regularity compared to various competing methods. Additionally, PPA metrics are tested using the commercial EDA tool Cadence Innovus, showing significant improvements.

**Strengths:**

In summary, this paper makes a substantial contribution to the field of chip design through its novel problem formulation and high-quality experimental methodology. The originality of integrating regularity into the RL framework and the thoroughness of the experimental validation are particularly noteworthy. This work advances the state of the art in macro placement.
1. Originality
- Novel Problem Definition: The paper introduces a new problem formulation by shifting the focus from using RL to place from scratch to refining existing macro placements. This innovative approach allows for more efficient use of state information and more precise reward signals.
- Integration of Regularity: The inclusion of regularity as a key metric in the RL framework is a novel contribution. Regularity is crucial for ensuring manufacturing consistency and performance but is often overlooked in existing methods.
2. Quality
- Robust Methodology: The paper employs a comprehensive experimental methodology, utilizing the ICCAD 2015 benchmark and commercial EDA tools (Cadence Innovus) for validation. The experiments are well-designed and include comparisons with multiple existing methods.
- Detailed Analysis: The paper includes a comprehensive analysis of the proposed method, along with ablation studies that validate the effectiveness of different components. The detailed evaluation and the provided code help to enhance the credibility of the results.
3. Significance
- Impact on Chip Design: The proposed Macro Regulator addresses significant limitations of existing RL methods in macro placement, such as training time, reward sparsity, and generalization capability. By improving placement quality and PPA metrics, this work has the potential to significantly impact the efficiency and effectiveness of chip design processes.

**Weaknesses:**

While the paper presents a significant advancement in the field of macro placement using reinforcement learning, there are areas for improvement. Expanding the benchmark scope, including real-world case studies, demonstrating generalization capabilities, and providing more detailed algorithmic descriptions would enhance the overall impact and robustness of the work. Addressing these weaknesses would make the paper's contributions even more compelling and practical for real-world applications.
1. Limited Benchmark Scope
- Additional Datasets Needed: While the paper utilizes the ICCAD 2015 benchmark and Cadence Innovus for validation, the scope of the benchmarks is somewhat limited. Including additional datasets or real-world applications could strengthen the evaluation and demonstrate the generalizability of the proposed method. For example, benchmarks from other well-known contests or industrial examples with different characteristics could provide a more comprehensive assessment.
2. Generalization to Different Chip Designs
- Generalization Capability: While the paper mentions the generalization capabilities of Macro Regulator, more detailed experiments and analysis on different types of chip designs (e.g., various sizes, complexities, and design constraints) would strengthen the claims. Demonstrating the method's adaptability to a broader range of scenarios would enhance its perceived robustness and applicability.
3. Clarity of Method Description
- Detailed Algorithmic Steps: While the paper is generally clear, some parts of the methodology could benefit from more detailed descriptions. For example, providing pseudo-code or more granular steps of the Macro Regulator algorithm would help readers better understand the implementation details and reproduce the results. Additionally, regularity is an important metric, but the paper provides limited explanation of this metric. It would be beneficial to supplement more information about its definition and calculation. Furthermore, a more detailed explanation of essential factors such as the states and actions in reinforcement learning would enhance the paper.

**Questions:**

In the abstract, the author states, "Our RL regulator can fine-tune placements from any method and enhance their quality." However, in the experimental section, it is mentioned, "For each chip, MaskRegulate uses DREAMPlace to obtain an initial macro placement result to be adjusted, which takes within a few minutes and has relatively low quality." There are two issues with this:
1. The claim of "from any method" is not entirely supported as DREAMPlace is essentially an analytical method, thus the experiments can only prove the ability to improve macro placement solutions from analytical methods, not from any other method.
2. The assertion of "relatively low quality" raises the question of whether the MaskRegulate method relies heavily on the quality of initial macro placement result. It would be worthwhile to explore the effect of optimizing from different quality macro placement solutions (such as higher but not the best macro placement result), rather than starting from relatively low-quality ones.

In summary, can the MaskRegulate method improve the quality of "any method" and "any quality of initial macro placement solution"?

**Limitations:**

Yes.

---

> ### Author Rebuttal · Authors · 2024-08-07
>
> Thank you for your detailed and valuable comments. Below please find our response.
>
> ### Q1 Limited benchmark scope.
>
> Thanks for your valuable comments. In our work, we use the ICCAD 2015 contest  as our benchmark, which is currently one of the largest open-source benchmarks that allows us to evaluate congestion, timing and other PPA metrics. To the best of our knowledge, ICCAD 2015 is the benchmark that closely reflects the current practices in the EDA industry. We agree with you that adding benchmark scopes would further strengthen our work. Thus, we add experiments on ISPD 2005 contest
> benchmark, which is also a popular benchmark in AI for chip design but does not have sufficient information for PPA evaluation. The results can be found in Table R1 in our PDF file.  For detailed settings and discussions, please refer to the following Q2. Thank you.
>
> ### Q2 Generalization ability
>
> Thank you for your valuable comments. In our paper, we have tested MaskRegulate's generalization ability in Table 2. We have now further tested the generalization on the ISPD 2005 benchmark by directly using the pre-trained models on superblue 1, 3, 4, and 5 (i.e., the same models in Table 2) of MaskPlace and MaskRegulate to place and regulate the eight chips. As shown in Table R1, MaskRegulate still outperforms MaskPlace in most cases, demonstrating our superior generalization ability and robustness. We will include this experiment in our revised paper. Thank you very much.
>
> ### Q3 Clarity of method description.
> Thank you for your suggestions. However, we cannot update the paper now due to the rules of rebuttal phase. We will carefully revise our paper according to your suggestions, including adding pseudo-code and introducing regularity and MDP in detail. Thank you very much.
>
> ### Q4 Can MaskRegulate improve the quality of any method and any quality of initial macro placement solution?
>
> Thank you for your valuable comments. Yes, MaskRegulate can be used to adjust any initial macro placement solution. According to your suggestions, we have conducted additional experiments to demonstrate this capability. We used the pre-trained model on superblue 1, 3, 4, and 5 (i.e., the same models in Table 2 and R1) to adjust different placement results obtained by MaskPlace, AutoDMP, and WireMask-EA. The results are shown in Table R2. MaskRegulate consistently improves regularity on all four unseen chips and enhances global HPWL on three chips. We will include this experiment in our revised paper. Thank you very much.
>
> **We hope that our response has addressed your concerns, but if we missed anything please let us know.**

---

> > ### Comment · Reviewer_NxAR · 2024-08-11
> > **Official Comment by Reviewer NxAR**
> >
> > Thank you very much for addressing my concerns. I will keep my score.

---

### Official Review · Reviewer_ChiY · 2024-07-22

**Soundness:** 3
**Presentation:** 3
**Contribution:** 2
**Rating:** 5
**Confidence:** 3

**Summary:**

This paper presents an application of the RL algorithm in chip design optimization. They formulate the chip design process as an MDP process to make decisions based on the current state of the chip macro arrangement so that they can apply PPO to optimize the policies. Compared with starting from scratch, they focus on the fine-tuning stage of the macros for denser reward signals. Their experiments in a public benchmark, the ICCAD 2015 benchmark show a great improvement in the chip design metrics.

**Strengths:**

According to the related work section, although there have been RL methods in this benchmark, this method improves previous methods by starting from macro placements instead of starting from scratch, which is a heuristic way of optimizing and simplifying the problem. This heuristic should be novel when combined with this class of methods.

The paper is technically solid in the problem formulation, experiments are sufficient with baselines, and reasonable ablation studies, and the results look good.

I like the visualization part of the paper.

**Weaknesses:**

Compared with MaskPlace [1], the contributions of this paper lie in the expertise or experiences in the expert chip design area instead of the machine learning area, as MaskPlace has already used PPO. One thing I am concerned about is that NeurIPS may not be a good venue to discuss this contribution.

[1] Lai, Y., Mu, Y., & Luo, P. (2022). Maskplace: Fast chip placement via reinforced visual representation learning. Advances in Neural Information Processing Systems, 35, 24019-24030.

**Questions:**

[1] When you discretize the map, it becomes a combinatorial problem. Did you compare the sample complexity of using RL vs using some combinatorial solver?
[2] PPO can handle continuous action space, is there a possibility of improvement using continuous action space?
[3] What will be the future directions for this work or is this problem solved by RL?

**Limitations:**

The authors have discussed the limitations of this work.

---

> ### Author Rebuttal · Authors · 2024-08-07
>
> Thank you for your valuable comments. Below please find our response.
>
> ### Q1 Compared with MaskPlace, the contributions of this paper lie in the expertise or experiences in the expert chip design area .... One thing I am concerned about is that NeurIPS may not be a good venue to discuss this contribution.
>
> Thanks for your comments. However, we may not agree with you respectfully.  Compared with MaskPlace, the main contributions of our work are two folds:
>
> 1. Problem formulation: MaskRegulate shifts the focus of RL from placing macros from scratch to refining existing macro placements, which allows for more effective learning from structured state and accurate reward information, significantly enhancing the learning efficiency.
>
> 2. Integration of regularity: We introduce regularity, a critical yet overlooked metric in chip design, into the RL training framework, which not only aligns with industry practice but also enhances the chip PPA quality.
>
> We have conducted comprehensive experiments to show the superior performance of MaskRegulate:
>
> - Our main experiments on the popular ICCAD 2015 benchmark have shown the significant improvements of MaskRegulate in PPA metrics, demonstrating the practical applicability and effectiveness of the RL regulator.
>
> - For the problem formulation, Table 6 in the paper has shown that Vanilla-MaskRegulate is better than MaskPlace. The only difference between them is the problem formulation, and all the other components (e.g., state and reward) are the same. Thus, the results in Table 6 clearly demonstrates our motivation, highlighting the advantages of our regulator problem formulation.
>
> - For the integration of regularity, Table 8 in the paper and Table R3 in the PDF file show that our proposed RegularMask can be integrated into other methods such as MaskPlace and WireMask-EA.
>
> Next, we will explain why NeurIPS is a good venue for our work. Recently, AI for chip design significantly expands the application domain of current AI technologies and enhance their impact [1]. In recent years, researchers have discovered that RL can assist chip design, as demonstrated in Google's Nature paper [2]. Since then, numerous top-tier venues (e.g., NeurIPS, ICML, and ICLR) have emerged with related research, aiming to further improve its effectiveness from different perspectives [3-9], just to name a few. Our work not only proposes a new placement paradigm but also introduces practical experience from the chip design field to guide the algorithm, which opens new possibilities for the application of RL in chip design. Therefore, we believe that a top-tier AI conference like NeurIPS is a proper venue for our work. Thank you.
>
> [1] Machine learning for electronic design automation: A survey. TODAES, 2021
>
> [2] A graph placement methodology for fast chip design. Nature, 2021.
>
> [3] On joint learning for solving placement and routing in chip design. NeurIPS, 2021.
>
> [4] MaskPlace: Fast chip placement via reinforced visual representation learning. NeurIPS, 2022.
>
> [5] Chipformer: Transferable chip placement via offline decision transformer. ICML, 2023.
>
> [6] Macro placement by wire-mask-guided black-box optimization. NeurIPS, 2023.
>
> [7] CircuitNet 2.0: An advanced dataset for promoting machine learning innovations in realistic chip design environment. ICLR, 2024.
>
> [8] Reinforcement learning within tree search for fast macro placement. ICML, 2024.
>
> [9] A hierarchical adaptive multi-task reinforcement learning framework for multiplier circuit design. ICML, 2024.
>
> ### Q2 Discussion about combinatorial solver.
>
> Good points! Thanks for your insightful comments.
> Placement is a very large-scale non-linear optimization problem. Using combinatorial solvers for placement has a long history [1-2], which belongs to analytical placement approach. They were initially used for placement of small-scale chips, but have gradually been replaced by more effective gradient-based approaches [3-4] in recent years. Currently, combinatorial solvers are typically used for small-scale and constrained problems, such as floorplanning [5].
>
> The state-of-the-art analytical placement method is DREAMPlace [4], which is an important baseline in our work. Compared to DREAMPlace, our RL regulator has many advantages, as demonstrated in our paper. According to your comments, we will introduce the history of analytical placers in detail to provide a comprehensive understanding of related techniques for readers without the placement background. Thank you for bringing our attention to this point.
>
> [1] Analytical approaches to the combinatorial optimization in linear placement problems. TCAD, 1989.
>
> [2] BonnPlace: Placement of leading-edge chips by advanced combinatorial algorithms. TCAD, 2008.
>
> [3] Replace: Advancing solution quality and routability validation in global placement. TCAD, 2018.
>
> [4] DREAMPlace: Deep learning toolkit-enabled GPU acceleration for modern VLSI placement. TCAD, 2021.
>
> [5] Global floorplanning via semidefinite programming. DAC, 2023.
>
> ### Q3 PPO can handle continuous action space, is there a possibility of improvement using continuous action space.
>
> Thanks for your suggestions. Due to the large action space of placement, previous studies propose to discretize the layout into grids, thus reducing the action space and improving learning efficiency. We agree that learning in the continuous space can enhance decision flexibility, but this also brings challenges such as low convergence rate and under-exploration. We will consider it in the future. Thank you.
>
> ### Q4 Future directions of RL
>
> Our main future work includes two aspects:
> 1. Further improving the generalization of the RL policy by using better architectures and training methods.
> 2. Utilizing some MO techniques (e.g., MORL) to obtain a set of Pareto-optimal placement results with different preferences across multiple objectives.
>
> **We hope that our response has addressed your concerns, but if we missed anything please let us know.**

---

> > ### Comment · Area_Chair_pxxv · 2024-08-11
> > **Dear reviewer, please read and respond to authors' rebuttal.**
> >
> > This paper has diverse reviews and it would benefit a lot to start a discussion to clarify any confusing points.
> >
> > Thanks!
> >
> > Your AC.

---

> > ### Comment · Reviewer_ChiY · 2024-08-11
> >
> > Thank authors for answering my questions, I have adjusted my ratings.
> > Follow-up Q2: I guess your RL method benefits from the reduction in the state/action space, and similarly, combinatorial methods can also benefit from this, was there any baseline run under the same action space or problem setting?

---

> > > ### Author Response · Authors · 2024-08-12
> > >
> > > Thank you for your feedback. We are pleased to hear that your concerns have been addressed and the score has been increased.
> > >
> > > Regarding the follow-up Q2, we agree that our proposed formulation of RL can also be used for combinatorial methods. To the best of our knowledge, no existing work has used combinatorial optimization under this setting. One may use some suitable combinatorial methods under our proposed RL regulator formulation and the reduction of search space to investigate if they can be improved. We will leave this for future work. Thank you for your insightful suggestions.

---

### Author Rebuttal · Authors · 2024-08-07

We are very grateful to the reviewers for carefully reviewing our paper and providing constructive comments and suggestions. We have revised the paper carefully according to the comments and suggestions, but we cannot upload the paper due to the NeurIPS rules. Our response to individual reviewers can be found in the personal replies, but we would also like to make a brief summary of revisions for your convenience.

Writing Enhancements:

- We correct typos and improve some method explanations to enhance overall readability.

Expanded Discussions:

- We incorporate a discussion on the combinatorial solver for macro placement.

- We explore the potential of using alternative approaches as macro regulators.

- We address the scenario where no feasible space is available and propose potential solutions.

Additional Experiments:

- We conduct generalization experiments using the ISPD 2005 benchmark to demonstrate the robustness of our method.

- We conduct generalization experiments to adjust various layouts obtained through different algorithms with different qualities.

- We implement and evaluate WireMask-BBO as a macro regulator using our proposed RegularMask.

- We include training convergence graphs to facilitate a comparative analysis of different RL-based approaches.

**We believe these revisions have significantly strengthened our paper and hope that your concerns have been addressed. We remain open to further feedback and are happy to provide additional clarification if needed.**

---

### Decision · Program_Chairs · 2024-09-25

**Decision:**

Accept (poster)

**Comment:**

This work proposes MaskRegulate that leverages PPO to find new chip designs, evaluates on standard benchmarks (ICCAD 2015) and shows better performance than existing baselines. The major contribution is to start from existing design and focus on refinement, and proposes novel intermediate metric like concept of regularity to improve the performance. Overall, all reviewers agree that the work is solid with thorough experiments.